# Post-Traumatic Meningitis Is a Diagnostic Challenging Time: A Systematic Review Focusing on Clinical and Pathological Features

**DOI:** 10.3390/ijms21114148

**Published:** 2020-06-10

**Authors:** Raffaele La Russa, Aniello Maiese, Nicola Di Fazio, Alessandra Morano, Carlo Di Bonaventura, Alessandra De Matteis, Valentina Fazio, Paola Frati, Vittorio Fineschi

**Affiliations:** 1Department of Anatomical, Histological, Forensic and Orthopaedic Sciences, Sapienza University of Rome, Viale Regina Elena 336, 00161 Rome (RM), Italy; raffaele.larussa@uniroma1.it (R.L.R.); nicola.difazio@uniroma1.it (N.D.F.); alessandra.dematteis@uniroma1.it (A.D.M.); valentina.fazio@uniroma1.it (V.F.); paola.frati@uniroma1.it (P.F.); 2IRCSS Neuromed Mediterranean Neurological Institute, Via Atinense 18, 86077 Pozzilli (IS), Italy; aniello.maiese@uniroma1.it; 3Department of Surgical Pathology, Medical, Molecular and Critical Area, Institute of Legal Medicine, University of Pisa, 56126 Pisa (PI), Italy; 4Neurology Unit, Department of Neurosciences, Mental Health, “Sapienza” University, 00161 Rome (RM), Italy; alessandra.morano@uniroma1.it (A.M.); carlo.dibonaventura@uniroma1.it (C.D.B.)

**Keywords:** post-traumatic meningitis, traumatic brain injury, diagnosis, epidemiology, risk factors, clinical data, laboratory data, therapeutic management, post-mortem diagnosis

## Abstract

Post-traumatic meningitis is a dreadful condition that presents additional challenges, in terms of both diagnosis and management, when compared with community-acquired cases. Post-traumatic meningitis refers to a meningeal infection causally related to a cranio-cerebral trauma, regardless of temporal proximity. The PICO (participants, intervention, control, and outcomes) question was as follows: “Is there an association between traumatic brain injury and post-traumatic meningitis?” The present systematic review was carried out according to the Preferred Reporting Items for Systematic Review (PRISMA) standards. Studies examining post-traumatic meningitis, paying particular attention to victims of traumatic brain injury, were included. Post-traumatic meningitis represents a high mortality disease. Diagnosis may be difficult both because clinical signs are nonspecific and blurred and because of the lack of pathognomonic laboratory markers. Moreover, these markers increase with a rather long latency, thus not allowing a prompt diagnosis, which could improve patients’ outcome. Among all the detectable clinical signs, the appearance of cranial cerebrospinal fluid (CSF) leakage (manifesting as rhinorrhea or otorrhea) should always arouse suspicion of meningitis. On one hand, microbiological exams on cerebrospinal fluid (CSF), which represent the gold standard for the diagnosis, require days to get reliable results. On the other hand, radiological exams, especially CT of the brain, could represent an alternative for early diagnosis. An update on these issues is certainly of interest to focus on possible predictors of survival and useful tools for prompt diagnosis.

## 1. Introduction

Despite the dramatic decrease in incidence and mortality rates observed over the past decades, especially in high-income countries [1], meningitis still represents a medical emergency, which requires early diagnosis and prompt treatment in order to avoid death or serious neurological sequelae. The occurrence of meningitis is mostly related to microbial agents, particularly bacteria (so-called pyogenic meningitis), but meningeal involvement may also be secondary to solid/hematological malignancies, autoimmune diseases (including the recently described IgG4-related hypertrophic pachymeningitis), and other causes of aseptic inflammation (e.g., drug-induced) [2]. Among several predisposing factors, traumatic brain injury (TBI) is of particular interest, not only owing to its prevalence and its potential intrinsic severity; indeed, post-traumatic meningitis is a dreadful condition that presents additional challenges, in terms of both diagnosis and management, when compared with community-acquired cases. Post-traumatic meningitis *stricto sensu* refers to a meningeal infection causally related to a cranio-cerebral trauma, regardless of temporal proximity; therefore, meningitis following neurosurgical procedures (e.g., craniotomy, in-dwelling catheter placement), even those performed because of severe TBI, will not be discussed here. Although the risk of meningitis is the highest within the first week after brain injury [3], it has been widely documented that meningeal infection might develop after several months, or even years, and a remote mild head trauma might be the only identifiable risk factor in otherwise unexplained cases [4], especially those recurring over time.

## 2. Methods

### 2.1. Eligibility Criteria

The present systematic review was carried out according to the Preferred Reporting Items for Systematic Review (PRISMA) standards [5]. We used an evidence-based model for framing a PICO question model (PICO: participants, intervention, control, and outcomes).

The question posed was the following: Is there an association between TBI and post-traumatic meningitis? (P) Participants: patients suffering to or death-related to post-traumatic meningitis. (I) Interventions: evaluation of clinical status in patients diagnosed with meningitis. (C) Control: healthy patients. (O) Outcome measures: clinical parameters and inflammatory and microbiological data in patients with post-traumatic meningitis. Studies examining post-traumatic meningitis, paying particular attention to victims of traumatic brain injury, were included. Study designs comprised case reports, case series, retrospective and prospective studies, letters to the editors, and reviews. The latter were downloaded to search their reference lists similarly to other papers, but yielded no other potentially eligible papers. The search was limited to human studies.

### 2.2. Search Criteria and Critical Appraisal

A systematic literature search and a critical appraisal of the collected studies were conducted. An electronic search of PubMed, Science Direct Scopus, and Excerpta Medica Database (EMBASE) from the inception of these databases to 15 April 2020 was performed.

Search terms were (“post-traumatic meningitis” OR “nosocomial meningitis”) AND (“community-acquired meningitis” OR “recurrent meningitis” OR “neurological infections in intensive care unit”) in title, abstract, and keywords. The bibliographies of all located papers were examined and cross-referenced for further relevant literature.

Methodological appraisal of each study was conducted according to the PRISMA standards, including evaluation of bias. Data collection entailed study selection and data extraction. Two researchers (R.L.R., P.F.) independently examined those papers whose title or abstract appeared to be relevant and selected the ones that analyzed post-traumatic meningitis. Disagreements concerning eligibility between the researchers were resolved by consensus process. No unpublished or grey literature was searched. Data extraction was performed by one investigator (A.M.) and verified by another investigator (V.F.). This study was exempt from institutional review board approval as it did not involve human subjects. Only papers in English were included in the search.

## 3. Results

### 3.1. Search Results and Included Studies

An appraisal based on titles and abstracts as well as a hand search of reference lists were carried out. The reference lists of all located articles were reviewed to detect still unidentified literature. This search identified 321 articles, which were then screened based on their abstract to identify their relevance in respect to the following:the human study so we excluded animal studies,clinical features,diagnosis,post-mortem findings,management of the study so we excluded methodologically incomplete design studies and those with no explicit mention about ethical issues.

With regard to ethical issues, we discarded the papers where the obtaining of informed consent on patients was not mentioned, where the approval of the ethics committee was not found and, finally, where the permission to publish personal data was not clearly explicit. Figure 1 illustrates our search strategy.

A total of 81 studies fulfilled the inclusion criteria (Table 1).

### 3.2. Risk of Bias

This systematic review has a number of strengths that include the amount and breadth of the studies, which span the globe; the hand search and scan of reference lists for the identification of all relevant studies; and a flowchart that describe in detail the study selection process. It must be noted that this review includes studies that were published in a time frame of 59 years; thus, despite our efforts to fairly evaluate the existing literature, study results should be interpreted taking into account that the accuracy of the clinical procedures, where reported, has changed over the years.

## 4. Discussion

### 4.1. Epidemiology and Risk Factors

In recent years, the incidence and morbidity of meningitis have been decreasing thanks to widespread vaccination programs, timely diagnosis, and more effective treatments. Nevertheless, this condition still represents a heavy burden, in terms of both global health and expenditures, with an estimated 2.82 million cases in 2016, the highest toll being paid in the so-called “meningitis belt”, including peri-Sahelian countries of sub-Saharan Africa [1]. In high-income areas, such as the United States and Western Europe, annual incidence is about 1–3 cases/100,000 people [6]. The above-mentioned public health policies have also produced remarkable changes in the distribution and prevalence of microbial agents commonly underlying the development of meningitis. More specifically, the number of cases related to Haemophilus influenzae type b and Neisseria meningitidis (which is responsible for seasonal outbreaks, especially affecting young adults) have dramatically dropped thanks to the introduction of conjugate vaccines, so that nowadays, Streptococcus pneumoniae is by far the most common causative organism of community-acquired meningitis among immunocompetent subjects [6], particularly adults over 50 years of age and children under 2. Apart from age-specificity, well-known risk factors for pneumococcal meningitis are immunocompromised states, either congenital (mainly defects of innate immunity) or acquired, such as diabetes, alcoholism, human immunodeficiency virus (HIV), cancer, splenectomy, and immunosuppressant medications [7]. Another common cause of meningitis in immunodeficient and elderly patients is Listeria monocytogenes, which is the third most frequent pathogen in adult cases of meningitis, after *S. pneumoniae* and meningococcus serogroup B [8]. Along with immunological factors, anatomical defects also represent a crucial predisposing condition for intracranial infections, as they produce a breach in the complex defense systems (e.g., skull, cerebrospinal fluid (CSF) spaces, blood–brain barrier, CSF–brain barrier) aimed to guarantee central nervous system (CNS) integrity and homeostasis. TBI, mainly owing to motor vehicle accidents and falls, is the leading cause of acquired anatomical defects, with an incidence of meningitis following moderate-to-severe head trauma estimated to be around 1.4% [9]. Among adults (6.2–12.14%) with trauma-related skull fractures, 20% present with fractures of the skull base, more frequently involving the anterior cerebral fossa (ACF) (47%), followed by middle fossa (22–37%), and posterior one (0.21–3%) (Figure 2 A–D) [10].

Basilar skull fractures are associated with higher risk of CSF leak, which occurs in 12–30% of these cases (compared with 2% of all head traumas), especially if ACF structures are damaged, where both bone and dura are thinner, and the latter is more adherent to the skull [11]. CSF leak is an ominous sign, as it depends on the presence of a direct communication between subarachnoid space and external environment, and thus facilitates the penetration of microbial agents (e.g., resident flora in nasopharyngeal mucosa), which might be responsible for meningitis and other intracranial infections. CSF leak may appear as rhinorrhea or, less frequently, as otorrhea, and is more commonly observed in frontal and sphenoid sinus involvement (30%), followed by ethmoid (15–19%) and cribriform plate (7.7%) fractures. Rhinorrhea and/or otorrhea might occur immediately after the trauma or be delayed, or even present as low-volume, intermittent discharges, which makes them difficult to detect. Although CSF leaks tend to resolve spontaneously after TBI, they might persist over days, increasing the risk of developing meningitis, especially when exceeding one week [12]. In cases of traumatic CSF fistulas, the rate of meningeal infection is about 9.1% in the first week, when the patient is generally hospitalized—so that a diagnosis of “nosocomial meningitis” is to be made; this proportion decreases to 8% per month for the first six months after TBI, and then drops to 8% per year [3]. As previously stated, meningitis might actually develop as a late consequence of trauma, after several months or years, and present as a periodic phenomenon. The so-called recurrent meningitis, which differs from relapsing or recrudescent ones, is generally defined as the occurrence of at least two episodes of meningeal infection sustained by different pathogens, or by the same organism, but with an interval superior to 3 weeks after the termination of antibiotic course for the first episode [13]. Recurrent meningitis accounts for 5–6% of adult community-acquired cases, with an estimated incidence of 0.12 cases/100,000/year [14,15]. In the review by Tebruegge and Curtis, more than a half of cases of recurrent meningitis were associated with anatomical defects, 47% of which were related to remote head trauma, accounting for an overall proportion of 28% [13]. In consideration of these findings, it not surprising that *S. pneumoniae* is by far the most common causative in recurrent meningitides, and in post-traumatic ones overall, with a detection rate in CSF cultures up to 85% in the paper by Adriani and colleagues [15]. Such evidence might be easily explained in light of the anatomical proximity between intracranial space and nasal cavities/paranasal sinuses, whose mucosa is normally colonized by microbial flora, including S. pneumoniae. In the case of penetrating brain injuries, other pathogens, generally present on the skin, such as S. aureus, coagulase-negative staphylococci (especially S. epidermidis), and facultative and aerobic gram-negative bacilli, might be involved in the genesis of meningitis [16,17,18,19,20,21,22,23,24,25].

### 4.2. Pathogenesis and Pathophysiology: A Brief Outline

Meningitis is defined as inflammation of the meningeal linings, in particular arachnoid and pia mater, and the subarachnoid space. Bacterial meningitis is the result of the penetration of pathogenic microorganisms within the subarachnoid space, which generates an inflammatory cascade with dramatic consequences. Microbial agents can reach meninges and contiguous spaces via different mechanisms, the most common being hematogenous spread from a distant infectious reservoir or after mucosal/skin penetration. This route of CNS invasion implies a complex process, including sequential steps: (1) microorganism adhesion to mucosal surfaces; (2) penetration, survival, and replication in the bloodstream; and (3) breaking through the blood–brain barrier (BBB) [6,26]. The main sites of bacterial penetration (either trans-cellular or para-cellular) are likely post-capillary venules and veins, which are supposed to be more permeable than arterial vessels. Within the subarachnoid space, there is a low concentration of complement components, which hampers opsonization and phagocytosis; therefore, once penetrated, bacteria can easily survive and proliferate. When microorganisms start to die (either because of nutrient deficiency or medications), their by-products tend to accumulate in the CSF, thus contributing to the activation of the inflammatory cascade [6,26].

Apart from hematogenous dissemination, there are other less common routes of CNS invasion potentially leading to meningitis; for instance, microorganisms may spread to meninges from contiguous infected sites, as happens in the case of otitis, mastoiditis, sinusitis, endophthalmitis, and so on. Moreover, bacterial penetration into the subarachnoid space might occur directly through a breach in the skull and adjacent soft tissues; this is the case of post-traumatic and iatrogenic meningitis (the latter being caused by neurosurgical procedures). In such circumstances, bacteria that physiologically colonize naso-pharyngeal epithelium and sinus mucosa are mainly responsible for the infectious process. Conversely, in the case of penetrating injuries, microorganisms normally present on the skin are likely to be involved [27,28,29,30].

The thorough revision of the complex pathophysiological mechanisms underlying meningitis is beyond the purpose of this review. However, some key concepts will be briefly highlighted.

The first consideration is that both host-related and pathogen-related factors play a crucial role in determining the onset and the severity of the disease. On the host’s side, either acquired (e.g., diabetes, HIV, cancer, medications) or congenital features (including single nucleotide polymorphisms, as recently documented) could contribute to immunological deficits and increase the susceptibility to meningitis, especially when involving the complement system and innate immunity in general. On the other side, not all microorganisms can cause meningitis; in fact, encapsulated bacteria (such as S. pneumonia and N. meningitidis) are more resistant to opsonization and phagocytosis, which facilitates their “escape” and survival within the bloodstream and the CSF, favoring the development of meningitis. Moreover, different serotypes of the same microorganism apparently have a different pathogenic potential, probably owing to peculiar molecular patterns (e.g., surface antigens) [6]. Besides, in order to survive to host’s defenses, bacteria can spontaneously change during the infection course, becoming more virulent and resistant to therapy [31,32,33].

Another important point that should be kept in mind is that meningitis-related brain injury is the result of not only the infection per se, but also the host’s inflammatory response. Indeed, when mediators of inflammation (e.g., tumor necrosis factor-α, interleukin-1, interleukin-6) are released in the CNS in response to infection, they can induce chemotactic and adhesion molecules that, along with bacterial components, lead to the influx and/or activation of leukocytes and glial cells. Polymorphonucleocytes and macrophages then produce tissue-damaging substances, such as proteases and oxidants, and trigger several processes (e.g., vasculature changes, cytotoxic/vasogenic edema) contributing to brain damage [26,27].

### 4.3. Clinical Manifestations

The clinical triad classically associated with pyogenic meningitis consists of fever, nuchal rigidity (“neck stiffness”), and alteration in mental status. Other frequent manifestations are headache and photophobia, whereas a variable proportion of patients might present with focal neurological deficits owing to the involvement of cranial nerves or brain parenchyma (so-called meningo-encephalitis). Some additional clinical features might also hint at the specific etiology; for instance, petechial rash is highly suggestive of meningococcal infection. However, it is widely acknowledged that typical clinical signs/symptoms may be far less apparent in children, especially in neonates, who generally present with nonspecific manifestations such as lethargia, irritability, poor feeding, respiratory distress, and hypo/hypertonia. Moreover, in adults themselves, the historical semiological triad might not be as sensitive as commonly thought; in the Dutch Cohort Study performed on 671 adult patients with community-acquired meningitis, only 44% of the population presented fever, neck stiffness, and altered awareness at disease onset [17]. When considering headache (reported by 87% of subjects) in addition to the other classical clinical features, the percentage of patients showing at least two of them raised to 85%. Regarding the specific etiology, the classical symptoms were more common during pneumococcal rather than meningococcal infection (58% vs 27%). *S. pneumoniae* was actually associated with a more severe disease course and worse outcome in comparison with other causative organisms. In accordance with these findings, a more severely impaired mental status and a higher incidence of seizures have been documented in pneumococcal meningitis, along with a greater proportion of focal neurological deficits related to the development of serious complications.

Notwithstanding, the sensitivity of clinical evaluation in the diagnostic process of bacterial meningitis can be greatly influenced by factors other than the patient’s age and the specific etiology, such as the subject’s immunological state and the use of specific medications (e.g., pain-killers). Besides, meningism might be related to several other neurological diseases apart from meningeal infection, first of all, subarachnoid hemorrhage (SAH).

Some semiological maneuvers should be mentioned because of their historical value, although their diagnostic accuracy has been proven to be very limited. The most popular signs among physicians include the following: (1) Kernig sign: when the subject is lying (or sitting) with flexed hip and knee, a resistance to passive knee extension can be appreciated. (2) Brudzinski signs: “nape of the neck sign”: neck flexion elicits the flexion of the ipsilateral hip and knee; “identical contralateral reflex”: the flexion of hip and knee on one side induces an identical contralateral response; “symphyseal sign”: the pressure applied over the pubic symphysis determines bilateral hip and knee flexion and leg abduction. (3) Jolt accentuation maneuver: the rapid horizontal rotation of the head (2–3 times/second) exacerbates headache [18]. These maneuvers have poor specificity as they reflect meningeal irritation, which might be related to causes other than infectious meningitis. Moreover, recent studies have demonstrated their sensitivity to be very low as well, ranging from 2% to 20%. In in this respect, jolt accentuation shows the highest sensitivity (up to 33% in subjects with moderate CSF pleocytosis); still, its diagnostic value is rather poor, and it might be risky in patients with recent cranio-cervical trauma [19,20].

As far as clinical features are concerned, post-traumatic meningitis does not differ from community-acquired cases; however, awareness impairment is generally more severe, and the clinical picture might be clouded by the presence of other brain lesions (e.g., intracranial bleeding, cerebral edema) or concomitant medical conditions, which makes the early recognition of meningitis even more challenging. In severe TBI (defined as Glasgow Coma Scale score < 8), general anesthesia and assisted ventilation are often required for patients’ management (as well as neuroprotection), which hampers the timely evaluation of changes in mental status and vital functions. Moreover, some compounds (e.g., opioids, neuromuscular blockers) might alter pupil diameter and reactivity, or muscle tone, sometimes masquerading impending neurological complications. Finally, infections are a common occurrence in critically ill patients, the most frequent being ventilator-associated pneumonia, urinary tract, and bloodstream infections, followed by meningitis/ventriculitis [21]; in this setting, it might be difficult to attribute fever specifically to intracranial infections, or to any infection, considering the chance of central hyperthermia in severely brain injured patients.

In subjects with recent head trauma, CSF leaks should be properly investigated: patients with rhinorrhea might report a salty taste (or sweet, at times) in their mouth, and “running nose”, that is, the occurrence of watery discharges, often experienced on standing, coughing, or sneezing [11]. Rhinorrhea is generally unilateral, occurring on the same side of the skull fracture, but it might seldom be paradoxically contralateral to the fracture, when the nostril of the affected side is obstructed by either a bone fragment or meningocele, or in the case of midline lesions [10]. When not immediately apparent, rhinorrhea could be elicited by flexing the subject’s neck or leaning the patient forward, what is usually called the “reservoir sign” [10]. However, CSF leaks might be of limited volume, delayed, or intermittent; therefore, their absence does not rule out the possibility of a skull fracture.

In conclusion, although pyogenic meningitis can be suspected when patients are admitted to the emergency department, especially when predisposing factors are identified (e.g., remote/recent head trauma, immunocompromised status), it cannot be diagnosed on clinical grounds only, and requires laboratory confirmation through CSF analysis.

### 4.4. Diagnostic Approach

#### 4.4.1. CSF Analysis

CSF examination is the gold standard for the diagnosis of infectious meningitis, whose etiological definition can be achieved only by CSF culture [6,8]. CSF analysis requires sample collection through lumbar puncture, a minimally invasive, easily manageable procedure. Lumbar puncture should be postponed to brain neuroimaging (CT scan) only for safety concerns, when clinical signs suggest the existence of conditions (e.g., space-occupying lesions, obstructive hydrocephalus) that might absolutely contraindicate lumbar puncture due to risk of intracranial hypertension with consequent brain herniation. This precaution is crucial in TBI patients, who might also have intracranial bleeding, skull or spine fractures with bone dislocations, or cerebral edema, and always undergo CT scan on admission. However, the need for a preliminary radiological exam must not hinder the timely introduction of antibiotic treatment, which should be empirically started even before lumbar puncture is performed, when deemed necessary. As for CSF analysis, findings typically observed in pyogenic meningitis include the following: reduced glucose level with low CSF/blood glucose ratio (<0.4); increased protein level (>2g/L); and polymorphonucleate pleocytosis, generally exceeding 1000 cells/mm^3^ [6]. Each of these findings is an independent predictor of bacterial meningitis, and at least one of them is detected in about 96% of patients with pyogenic meningeal infection. Nevertheless, the extent of CSF abnormalities is closely related to the subject’s age, immunological state (e.g., cancer patients have less marked CSF pleocytosis, as well as those with ongoing septic shock), causative agent, and concomitant medications. CSF culture is essential for organism identification and has a high sensitivity (ranging from 60% to 96%, depending on the specific organism), although its diagnostic accuracy is affected by antibiotic pre-treatment [34]. CSF Gram stains could play a crucial role in cases with negative cultures thanks to its high specificity, which makes it suitable for directing prompt therapeutic choices. Once again, etiological factors and antibiotic treatment could lower the diagnostic yield of this technique. Ancillary tests, such as polymerase chain reaction (PCR) on CSF, might be of help in double-negative cases, whereas latex agglutination test is of limited value [35].

#### 4.4.2. Surrogate Serum Markers

When CSF cannot be obtained (e.g., owing to the high risk of intracranial hypertension, or for difficulties in performing the lumbar puncture related to the spine anatomy, prior back surgery, obesity, and so on), surrogate serum markers supporting the clinical suspicion of meningitis are needed. In this regard, C-reactive protein (CRP) has recently gathered considerable attention, and even more so, procalcitonin (PCT). The latter is a 116-aminoacid protein physiologically synthesized by thyroid C cells as a precursor of calcitonin, and its level raises during sepsis thanks to increased extra-thyroidal production (e.g., in adipocytes, hepatocytes, different parenchymal cells), probably induced by bacterial products (i.e., endotoxin) and pro-inflammatory cytokines (e.g., tumor necrosis factor-α, interleukin-6) [36,37]. A recent meta-analysis demonstrated its high accuracy in distinguishing between viral and bacterial etiology in cases of suspected meningitis, proving PCT to be superior to all CSF parameters (i.e., low glucose level, high protein concentration, leukocytes) [37]. Moreover, this marker has been demonstrated to rise 4 h after the onset of meningitis, peak at 6 h, and persist over 24 h, thus representing a useful tool in the early phase of the disease. However, PCT is not useful for etiological definition, nor is it specific for intracranial infections. Besides, as previously discussed, infections and sepsis are common complications in critically ill patients, like those with severe TBI hospitalized in intensive care unit (ICU); therefore, increased levels of PCT (>10 ng/L), however suggestive, cannot specifically point to meningitis. Conversely, blood cultures might be a useful tool for organism isolation when CSF samples are not available or CSF cultural tests are negative [36]. Finally, several works have also investigated the use of serum leukocytosis (common cut-off 10 × 10^3^/μL) for the diagnosis of pyogenic meningeal involvement; so far, the bulk of literature data has proved the poor sensitivity and specificity of this parameter, whose positive predictive value is about 13–56% [38].

#### 4.4.3. The Role of Neuroimaging in Meningitis 

Neuroimaging studies are not crucial for meningitis diagnosis, apart from their role in identifying patients at risk for brain herniation, who should not undergo lumbar puncture. In the early phase, abnormal findings are seldom detected by CT scan; in the Dutch cohort study on community-acquired meningitis, CT was performed on admission in 71% of subjects, and was unremarkable in almost 66% of them [17]. Contrast-CT might identify meningeal enhancement in a minor proportion of cases later on in the disease course. Magnetic resonance imaging (MRI) is not immediately required in uncomplicated meningitides, although gadolinium-enhanced MRI (Gd-MRI) can clearly document the meningeal involvement (Figure 3A–B).

However, Gd-MRI is crucial for detection of harmful complications such as cerebral ischemia, sinus thrombosis, brain abscess, hydrocephalus, and cerebral edema. Therefore, it should be promptly performed in patients with new-onset focal neurological deficits and seizures, or other signs of clinical deterioration [38,39]. In subjects presenting with TBI, neuroimaging studies are mandatory to detect skull fractures, intracranial bleeding, cerebral edema, and diffuse axonal injury. Brain imaging also play a pivotal role in localizing CSF fistulas, even when no CSF leak is clinically evident (because of meningocele, herniation, bone fragments, or blood clots temporarily obstructing the dural tear) [10]. The early identification of CSF fistulas is strategic to plan therapeutic approaches and estimate the risk of developing post-traumatic meningitis. With this aim, several advanced techniques have been applied, such as high-resolution CT (HRCT), which uses 1–2 mm sections in both the coronal and axial planes to obtain detailed images of the bone structures, or MRI cisternography, a non-invasive technique where T2-weighted images with fat suppression and image reversal allow to highlight CSF and brain parenchyma, despite poor bone definition [10]. However, the current radiological gold standard for identification of the exact site of CSF leak implies the intrathecal injection of a detectable dye, such as fluoresceine (used off-label) or radiopaque contrast followed by CT scanning (so-called CT cisternography), or even (less common) a radioactive substance, used in radionuclide cisternograms [11]. Despite their good sensitivity, these techniques have serious potential complications, including aseptic meningitis due to intrathecal injections of chemical compounds.

#### 4.4.4. Novel Diagnostic Tools

Finally, when CSF leak is suspected, a novel interesting, non-invasive approach consists of the (qualitative) identification in nasal discharges of β2-transferrin, a desialidated isoform of transferrin almost exclusively detected in CSF, with sensitivity and specificity about 99% and 97%, respectively [10,11]. The drawbacks of this assay are the time (24–48 h) needed to obtain the results and the evidence that β2-transferrin can be also found in aqueous humor and in the sera from patients with alcohol-related cirrhosis. A less expensive and much quicker approach is the quantitative measure of the levels of β-trace protein, a protein synthesized by meninges and choroid plexus, which has 100% sensitivity in the case of active rhinorrhea, but unfortunately is also increased in the serum of subjects with kidney failure [11].

### 4.5. Therapeutic Management

#### 4.5.1. Antibiotic and Steroid Treatment in Post-Traumatic Meningitis

Antimicrobial treatment is the cornerstone of the therapeutic management of all pyogenic meningitides; antibiotics should be started as early as possible, even before the isolation of the causative microorganism through CSF culture (which takes 24–48 h), in order to minimize the risk of death and permanent sequelae. Indeed, therapeutic delay (mainly related to brain imaging) is strongly associated with poor outcome [40,41,42]. Before the specific pathogen is identified, empirical antimicrobial treatment should be decided based on the patient’s age, known risk factors (e.g., immunodeficiency), and local epidemiological data about pneumococcal resistance. As already stated, *S. pneumoniae* is the most common causative microorganism in adult subjects with both community-acquired and post-traumatic meningitis; therefore, penicillin and third-generation cephalosporins represent the main compounds used in empirical therapy. However, pneumococcal reduced susceptibility to third-generation cephalosporins has become an increasingly worrying phenomenon, so that, in regions with expected high resistance rates, either vancomycin or rifampicin should be added [8]. Moreover, considering Listeria resistance to cephalosporins, all patients above the age of 60 should receive amoxicillin or ampicillin, as well as immunocompromised subjects. After the identification of the causative pathogen, the patient should be switched to a targeted treatment (see specific recommendations on this topic). Antibiotics could be administered as either continuous infusion or repeated boli, without proven differences in patients’ outcome. Conversely, therapy duration is still a matter of debate, but in culture-negative patients, the antibiotic course should last at least two weeks [8]. Corticosteroids have been lately evaluated as adjunctive therapy for acute bacterial meningitis, with the aim to reduce neuronal damage due to infection-related inflammatory mechanisms. A recent Cochrane review, including 25 studies on a total of 4121 pediatric and adult subjects, demonstrated that, exclusively in high-income countries, early steroid treatment is effective in preventing hearing loss and short-term neurological sequelae [43,44]. No effect on mortality was observed; however, subgroup analysis revealed a reduction in mortality rates among patients with S. pneumoniae. On the basis of the findings of the most relevant included trials, corticosteroid treatment, started before or within 4 h after the first antibiotic dose, is currently recommended in cases of *S. pneumoniae* and H. influenzae meningitis [8,45]. At present, dexamethasone is the compound of choice, and should be administered at low dose (10 mg) every 6 h for a duration of 4 days. However, caution is warranted in the case of septic shock and recent cranio-cerebral trauma. As for adjunctive osmotic therapy, glycerol is the only agent that has been evaluated in acute bacterial meningitis; similar to corticosteroids, it was demonstrated to be potentially effective in reducing neurological sequelae, but not mortality, according to the cumulative findings of five trails gathering 1451 individuals [46,47,48].

#### 4.5.2. Antibiotic Prophylaxis after TBI: A Matter of Debate

Another controversial issue in the therapeutic management of patients with meningitis is the need for antibiotic prophylaxis in subjects with head trauma, especially those presenting with severe TBI. Indeed, despite the relatively low incidence of meningeal involvement among patients with recent trauma, when well-known risk factors for meningitis (such as basilar skull fractures or persistent CSF leaks) can be identified, chemoprophylaxis seems a prudent choice, and is often used in clinical practice. Nevertheless, the potential adverse events related to antibiotics, and the risk of increased microbial resistance, require careful considerations. A recent Cochrane meta-analysis has attempted to cast light on such debated topic; in fact, the available evidence did not allow determining the actual utility of antibiotic prophylaxis in patients with basilar skull fractures, so that prophylactic treatment is not currently indicated [48,49,50,51]. More specifically, the meta-analysis included five randomized controlled trials (RCTs) performed on a total of 208 subjects; no differences—in terms of meningitis rate, meningitis-related mortality, and all cause-mortality—were found between patients receiving any antibiotics and those receiving placebo or no treatment at all, regardless of the presence of CSF leak (when specified). These findings were also confirmed after considering data from non-randomized trials for analysis. Another interesting question is whether subjects with post-traumatic meningitis might benefit from pneumococcal vaccination, considering the risk of recurrence. Despite the lack of RCTs properly investigating vaccination utility in recurrence prevention, the European Society for Clinical Microbiology and Infectious Diseases currently recommends vaccination in all patients with history of pneumococcal meningitis [8,52]. 

#### 4.5.3. The Management of CSF Leaks

Finally, in subjects with recent TBI, CSF leaks require careful monitoring and particular consideration. Rhinorrhea and otorrhea generally tend to resolve spontaneously over a few days, and a conservative approach is thus preferred. The patient should lie in bed with 30° head elevation and should be advised against coughing, sneezing, nose blowing, performing Valsalva maneuver, and so on [53,54,55,56,57,58]. Blood pressure should be strictly controlled in order to avoid further leak and allow the dural tear to heal. When CSF leak persists beyond one week, a surgical repair of the anatomical breach could be attempted, using different approaches (e.g., transcranial, extra-cranial, trans-nasal, endoscopic endonasal, and so on) whose discussion is beyond the purpose of this paper.

### 4.6. Outcome and Neurological Complications

In spite of recent improvements in terms of diagnostic accuracy and therapeutic management, bacterial meningitis still represents a serious infectious disease worldwide, with high mortality and morbidity especially in low-income countries, where the impact on quality of life and health economy is not negligible. The Global Burden of Diseases, Injuries, and Risk Factors (GBD) 2016 study showed a decrease in overall mortality rates by 21% from 1990 to 2016 [1]. In 1990, the main cause of meningitis deaths was N. meningitidis, but thanks to the introduction of vaccine programs and chemoprophylaxis, the mortality related to meningococcal infection has dropped. In the already mentioned Dutch cohort study on community-acquired meningitis in adult subjects, the overall mortality rate was about 20%, and it raised to 30% when considering pneumococcal infections (compared with 7% of meningococcal ones) [17]. Indeed, it is widely acknowledged that *S. pneumoniae* is associated with more severe disease course, poorer outcome, and more frequent long-term complications. No specific data about fatalities in post-traumatic meningitis are currently available, which can be easily explained considering its low incidence and difficult diagnosis, and the great impact of concomitant TBI-related lesions on outcome. Interestingly, recurrent meningitis has a better prognosis than other forms of meningeal infections, probably because affected patients learn to recognize early symptoms and seek medical attention [13].

A remarkable proportion of subjects suffering from meningitis struggle with neurological sequelae; these are generally related to the development of serious complications during the disease course, which might lead to new-onset focal neurological deficits, seizures, and sudden worsening of the patient’s mental status. One of the main complications observed in meningitis is cerebral ischemia, documented in up to 25% of adult patients; cerebral infarction might be caused by either arterial or venous sinus thrombosis, which likely depends on abnormal coagulation mechanisms, vasculitis, or vasospasm induced by the host’s inflammatory response [59,60,61,62]. Nevertheless, heart ultrasound should be performed in order to rule out cardio-embolism as a possible cause of cerebral ischemia.

Meningeal infection could also extend, involving subdural space and cerebral parenchyma, leading to difficult-to-treat conditions such as subdural empyema and brain abscess, which sometimes require surgical treatment. Moreover, hydrocephalus might also develop as a consequence of meningitis; in such cases, it is often non-resorptive (communicating), related to impaired CSF re-absorption by subarachnoid granulations owing to exceeding protein and cell content in CSF, and could spontaneously resolve [63,64,65,66,67,68]. However, obstructive hydrocephalus might also occur, especially in children, contributing to intracranial hypertension. This is an ominous condition, often following brain edema, which might lead to brain herniation and death. It also represents an absolute contraindication to lumbar puncture, hampering the diagnosis of meningitis. Subjects with severe TBI are even more at risk of intracranial hypertension because of possible concomitant lesions such as intracranial bleeding and diffuse axonal injury; therefore, invasive intracranial pressure (ICP) monitoring might be indicated in ICU patients.

Seizures might occur during the acute phase of meningitis (reportedly, in 9–34% of subjects), and require emergency treatment to avoid cardio-respiratory complications and further brain injury. However, they might also present later on, and mark the onset of chronic epilepsy attributed to remote meningitis (and related brain lesions). Anti-epileptic prophylaxis appears effective only in preventing early seizures (i.e., those occurring during the first week), whereas it does not influence the development of late seizures and epilepsy; however, RCTs dealing with this issue are lacking [8]. 

Another common early complication in patients surviving meningitis is sensorineural hearing impairment (reported by 22% of adults), which is thought to depend on direct spreading of bacteria and pro-inflammatory molecules through sub-arachnoid space to cochlea, with consequent CSF–labyrinth barrier break and nerve damage [69,70,71,72,73,74,75,76,77]. Apart from the recent introduction of dexamethasone as adjunctive therapy, which seems beneficial on hearing impairment, an early careful hearing evaluation is warranted, in order to detect the problem and prevent cochlear ossification, which could jeopardize the utility of cochlear implants.

Finally, cognitive deficits have been equally observed in patients surviving pneumococcal and meningococcal infections, with rates up to 30%. The early recognition of cognitive impairment is particularly relevant for children, who might benefit from educational support.

Recently, at molecular level, new diagnostic and therapeutic approaches have been proposed. Rapidly after TBI, it provokes the activation of the endothelium and a neuroinflammatory response, as demonstrated by the recruitment and up-regulation of cytokines, chemokines, neutrophils, and other pro-inflammatory mediators [78,79,80]. Neuropeptides such as calcitonin gene-related peptide (CGRP), neurokinin A (NKA) and B (NKB), and substance P (SP) are central to the development of neurogenic inflammation [81,82,83]. Looking for specific biomarkers, SP is the most potent initiator of neurogenic inflammation, with CGRP is able to further potentiate the effects of SP. The mechanism of the release of interleukin (IL)-1β, tumor necrosis factor (TNF), transforming growth factor beta (TGFβ), and monocyte chemoattractant protein-1 (MCP-1) is linked to the innate resident immune cells that are subjected to the activation of albumin, entering the brain through transcytosis, which also causes the release of matrix metalloproteinase (MMP) [80,84,85]. The inhibition of MMPs and A disintegrin and metalloproteinases (ADAMs), central regulators and mediators, appears to be a promising therapeutic intervention to prevent the development of neuronal damage during neuroinflammation [86,87]. Ubiquitin C-terminal hydrolaseL1 (UCHL1) represents an important TBI biomarker to support therapeutic efficacy of drugs [88]. Neurofilaments (NFs) are fundamental players in structural support and regulating axon diameter. The heavy neurofilament (pNF-H) subunit has been claimed as sensitive marker of axonal injury following TBI [89]. Neuron specific enolase (NSE) is a glycolytic enzyme released into the extracellular space under pathological conditions during cell destruction. Levels are high in the CSF, but the serum concentration depends on the state of the BBB [90,91]. In pediatric TBI, NSE has been correlated with the Glasgow Outcome Score (GOS) [92]. The hypothesis is about a protective role against blood–brain barrier (BBB) damage, inhibiting neuronal apoptosis and promoting angiogenesis [93,94].

### 4.7. Forensic Pathology Approach in Cases of Post-Traumatic Meningitis-Related Death

In the evaluation of liability features in the case of decease following post-traumatic meningitis, there are two main forensic pathology issues: the identification of meningitis as cause of decease, and the need for establishing the correlation between meningitis and the traumatic event. Bacterial meningitis death is rarely attributed to the actual event that caused death. Fatal complications from primary infection, meningitis, present many differences in the complications that cause death, so that determining the clinical cause of death is essential for evaluating causal correlation and new therapeutic strategies [95].

A correct methodological approach plays a fundamental role within the forensic pathology evaluation in the case of post-traumatic meningitis-related death.

These procedural criteria should include the following steps:-Complete autoptic examination;-Liquor sampling and microbiological investigations;-Brain in toto removal and formalin embedding;-Sampling of the intracranial structures;-Histological and immunohistochemical specimen’s microscopic evaluation.

#### 4.7.1. Autopsy

The consolidated forensic pathology protocols state that, even in cases of post-traumatic meningitis, the execution of the autoptic examination cannot be separated from a preliminary and scrupulous external examination of the corpse. Special attention should be paid regarding the evaluation of both direct and indirect traumatism signs, especially referring to the cranial region. Apart from bruises, lacerated wounds, or skin abrasions assessment, the presence of otorrhea or rhinorrhoea should be searched, as those events are associated to a facilitate liquor penetration and contamination from pathogen microorganism, with an incidence as great as 20% [96].

In light of the necessity to evaluate traumatic lesions that occurred within the facial region, anterior scalp flap and facial skin overturning should be exercised, until the exposure of nasal bones. This allows to directly evaluate the periorbital region as well as zygomatic and nasal bones. Then, skull bone is opened, allowing to inspect the underlying meningeal structures, which are of special interest in the infectious disease investigated, before incision and removal of the dura matter.

At this point, preliminary to the brain in toto removal, liquor should be collected in order to permit microbiological analysis. This is obtained by lateral ventriculi in situ incision, making two semicircular cuts laterally to cingulated gyrus (Figure 4A–B).

Hence, the brain may be removed and inspected in order to evaluate the presence of haemorrhagic lesion or hematomas, which should be adequately described. At the end, a careful inspection of the skull base bone should be carried out because fractures in this area represent the preferred site of entrance to the endocranial cavity.

#### 4.7.2. Tissues’ Sampling and Histological Examination 

The brain, which has been previously removed during the autopsy, must be embedded in formalin and afterwards re-examined before sampling. The macroscopic exam of the structure must be conducted after sectioning using different typology of cuts, depending on the kind of lesivity detected (Figure 5A–B).

Standard sampling is carried out with the addiction of sampling within the previously described injured areas. It follows the microscopic analysis of the samples where, at the hematoxylin-eosin staining, immunohistochemical staining can be associated in order to obtain a correct dating of the event. Then, microbiological exams on liquor permit to complete the diagnosis of the infectious process, whether this was not made when the patient was still alive.

#### 4.7.3. Forensic Pathology Assessment

Regarding the assessment of a causal correlation between the traumatic event and the meningeal infection, this may result to be extremely complex. In fact, as described in the scientific literature, the time-lapse between trauma and clinical signs of infectious disease result can be extremely variable, from a minimum of a few hours to a maximum time of several years [96,97,98,99].

Concerning dating of both the traumatic event and the infective disease, this can be integrated with the execution of immunohistochemical investigations aimed to detect specific molecular markers. While evaluating brain injury several proteins such as Aquaporin4 (AQP4), CD58, CD68, and antibodies anti-glial fibrillary acid protein (GFAP) should be researched. Previous studies identified that these molecular markers expression show a temporal correlation with the event in the case of traumatic brain injury [96] (Table 2).

It is also possible to perform immunohistochemical studies for the post-mortem diagnosis of sepsis related to meningitis [97,98]. For the pathologist, the historical immunohistochemical markers used (β-APP, GFAP) will continue to play an important role in the diagnostic evaluation process; the best understanding of the molecular mechanisms involved in the pathogenesis of the resulting damage to a TBI, however, as well as the understanding of oxidative stress leading to apoptosis, will assume greater importance, as the potential to allow for a more precise diagnosis in time compared with classic markers (Figure 6).

## 5. Conclusions

Post-traumatic meningitis represents a high mortality disease. Proper survival of the clinical course of the brain traumatic injury today still represents a great challenge for the clinicians, with the considerable implications that proper dating takes in assessing the clinical case. Diagnosis, which is often made post-mortem, may be difficult either because clinical signs are nonspecific and blurred or because of the lack of pathognomonic laboratory markers. Moreover, these markers increase with a rather long latency, thus not allowing a prompt diagnosis, which could improve patients’ outcome.

Among all the detectable clinical signs, the appearance of cranial CSF leakage (manifesting as rhinorrhea or otorrhea) should always arouse suspicion of meningitis. In fact, even though CSF leaks tend to resolve spontaneously after TBI, they might persist over days, increasing the risk of developing meningitis, especially when exceeding one week.

On one hand, microbiological exams on CSF, which represent the gold standard for the diagnosis, require days to get reliable results. For this purpose, CRP and PCT have been evaluated as early meningitis markers, being sensitive, but not specific enough. On the other hand, radiological exams, especially CT of the brain, could represent an alternative for early diagnosis. Indeed, CT can detect those alterations related to trauma, but is not specific in assessing the infectious disease.

Difficulty in the use of inflammatory markers such as immunohistochemical markers of TBI in early stage is the result of the overlap of secondary changes after a traumatic event (es. hypoxia, edema) that can lead to difficulty with errors in estimating the survival time [99]. Research to identify the miRNA expression profile secreted by astrocytes in response to acute neuroinflammatory stress has been performed recently [100]. Five miRNAs released by astrocytes under IL-1β-stimulated acute neuroinflammatory stress have been identified and extensively characterized [96]. These miRNAs may serve as potential biomarkers of CNS inflammation [101,102,103,104]. An increased expression of miR-21 after traumatic brain injury (TBI) has been described. Data indicate that miR-21, miR-92, and miR-16 have a high predictive power in discriminating trauma brain cases from controls and could represent promising biomarkers as strong predictor of survival, and a useful tool for postmortem diagnosis of traumatic brain injury [104]. More recently, miR-124-3p, miR-219a-5p, miR-9-5p, miR-9-3p, miR-137, and miR-128-3p exhibited dramatically greater brain specificity as blood biomarkers of TBI [105].

## Figures and Tables

**Figure 1 ijms-21-04148-f001:**
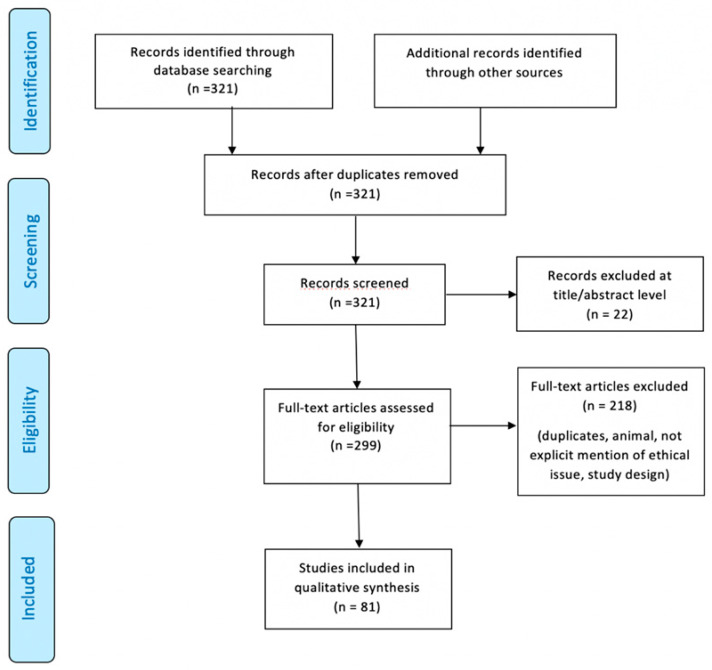
Preferred Reporting Items for Systematic Review (PRISMA) flow chart—search strategy. Study designs comprised case reports, case series, retrospective and prospective studies, letters to the editors, and reviews. An appraisal based on titles and abstracts as well as a hand search of reference lists were carried out. The reference lists of all located articles were reviewed to detect still unidentified literature. A total of 81 studies fulfilled the inclusion criteria.

**Figure 2 ijms-21-04148-f002:**
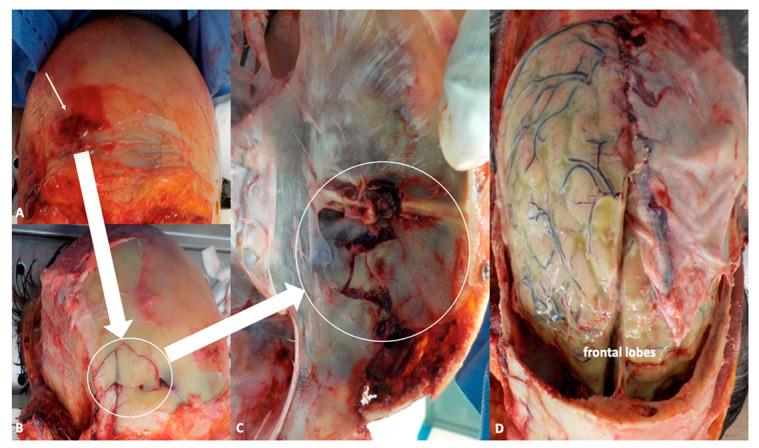
(**A**,**B**) Autoptical findings: exposure of the skull by facial skin overturning. This technique allows to evaluate the periorbital region as well as zygomatic and nasal bones (white arrow indicates fractures of the frontal region). (**C**) Fractures of the anterior fossa are easily detectable after removal of the brain (white arrow), while inspecting the basal skull. (**D**) In situ inspection of the brain. The leptomeninges shows frank green color owing to the presence of purulent material stratified under the meninges.

**Figure 3 ijms-21-04148-f003:**
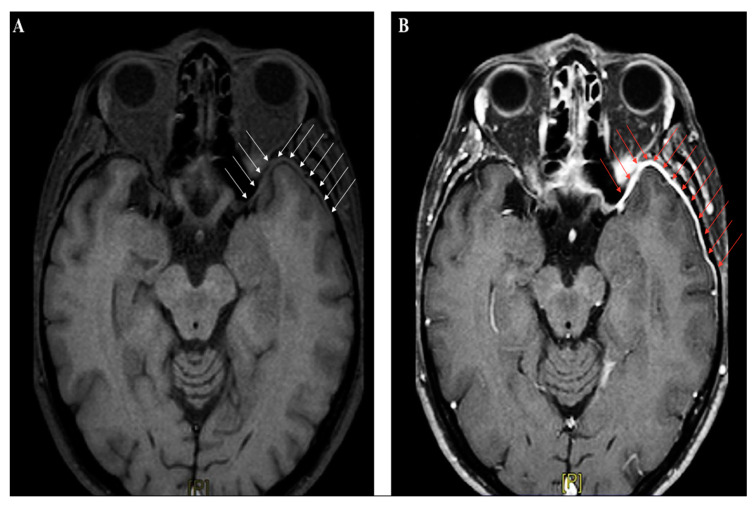
Brain magnetic resonance imaging (MRI) (axial section, T1-weighted images) performed in a patient with recurrent headache and diplopia, and a history of remote head trauma; a definite thickening of the meningeal lining over the left anterior temporal lobe (white arrow panel **A**), with following contrast enhancement (red arrow panel **B**), can be clearly identified.

**Figure 4 ijms-21-04148-f004:**
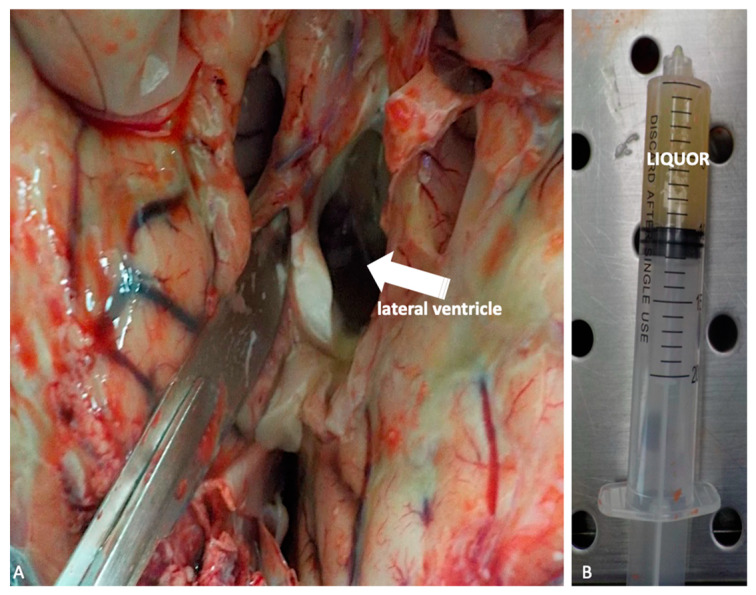
(**A**,**B**) Liquor sampling by lateral ventriculi in situ (white arrow) incision intended for microbiological analysis.

**Figure 5 ijms-21-04148-f005:**
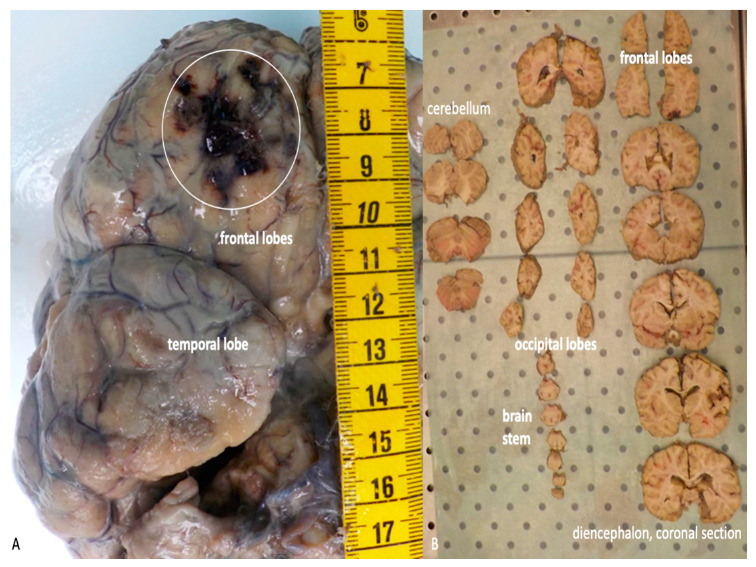
(**A**) Macroscopic examination of the brain after formalin embedding. A traumatic contusion area at the level of the frontal lobe is evident (white circle). (**B**) Serial coronal sections of the brain.

**Figure 6 ijms-21-04148-f006:**
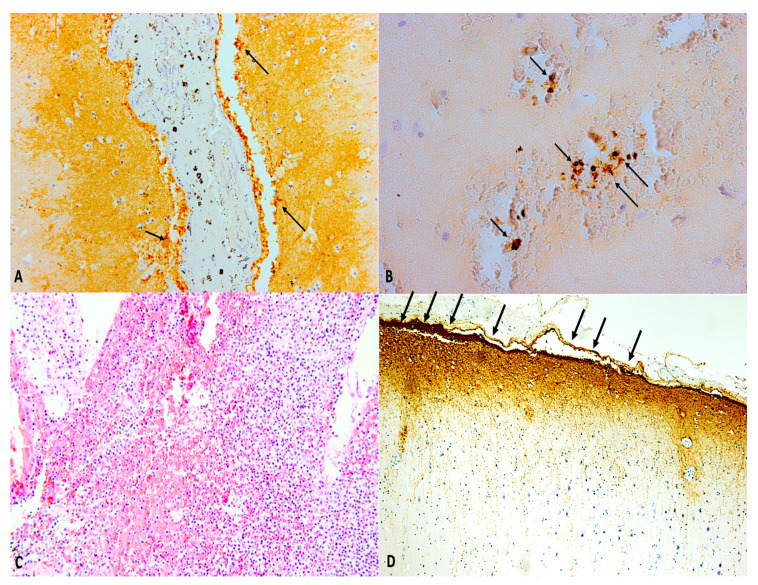
(**A**,**B**) Immunohistochemistry of both brain and meninges allows dating of the traumatic event post traumatic brain injury. (**A**) Aquaporin-4 (AQP4) and (**B**) hypoxia induced factor-1α (HIF-1α) immunopositivity (black arrows) in post-traumatic meningitis after traumatic brain injury (30-day survival) (×40 and ×100, respectively). (**C**) Post-traumatic meningitis: histological staining of the brain and meningeal tissues showing strong inflammatory response (hematoxylin and eosin (H&E), ×60). (**D**) Glial fibrillary acid protein (GFAP) showed its highest expression in post-traumatic meningitis after traumatic brain injury, a definite thickening of the meningeal lining (black arrows) is clearly observable (30-day survival) (×40).

**Table 1 ijms-21-04148-t001:** A total of 81 studies fulfilled the inclusion criteria. Details of all studies included in this systematic review.

Authors	Year	Description
Vos, T.	2017	The Global Burden of Diseases, Injuries, and Risk Factors Study 2016 (GBD 2016) provides a comprehensive assessment of prevalence, incidence, and years lived with disability (YLDs) for 328 causes in 195 countries and territories from 1990 to 2016. The decrease in death rates since 1990 for most causes has not been matched by a similar decline in age-standardised YLD rates
Yelehe-Okouma, M.	2018	Aseptic meningitis associates a typical clinical picture of meningitis with the absence of bacterial or fungal material in the cerebrospinal fluid. About 62 references were found, and only 18 were selected based on their type (case reports, review) and relevance (study characteristics, quality, and accuracy). DIAM remains a diagnosis of elimination.
Eljamel, M. S.	1990	The management of acute traumatic cerebrospinal fluid (CSF) fistulae is still a matter of debate and hinges on what is perceived to be the risk of subsequent intracranial infection. They have thus carried out a retrospective analysis of 160 cases of traumatic CSF leaks. The overall incidence of meningitis in this group before surgical dural repair was 30.6% (49/160), the cumulative risk exceeded 85% at 10-year follow-up, and the meningitis was fatal in 4.1% (2/49).
Matschke, J.	2001	Six cases of post-traumatic meningitis as the cause of death from the archives of the Institute of Legal Medicine in Hamburg are presented. There were all males; age varying between 24 and 90 years (mean 58 years); range of the interval between original trauma and beginning of symptoms was 2 days up to 8 years; in 50% of the cases, meningeal swabs yielded Streptococcus pneumoniae.
Van de Beek, D.	2006	In this review, they summarize the current concepts of the initial approach to the treatment of adults with bacterial meningitis, highlighting adjunctive dexamethasone therapy, and focusing on the management of neurologic complications.
Adriani, K. S.	2015	Bacterial meningitis is a life-threatening infectious disease with high mortality and disability rates, despite availability of antibiotics and adjunctive therapy with dexamethasone. Several risk factors and predisposing conditions have been identified that increase susceptibility to bacterial meningitis: immunodeficiency, host genetic factors, or anatomical defects of the natural barriers of the central nervous system.
Van de Beek, D.	2016	This is the ESCMID guideline for diagnosis and treatment of acute bacterial meningitis.
Van de Beek, D.	2010	Nosocomial bacterial meningitis may result from invasive procedures; complicated head trauma; or, in rare cases, metastatic infection in patients with hospital-acquired bacteremia. These cases of meningitis are caused by a different spectrum of microorganisms.
Phang, S. Y.	2016	Using an algorithm, the authors studied the most suitable management of cerebrospinal fluid losses after skull base fracture. The results obtained revealed some many unresolved questions, which will need further studies.
Prosser, J. D.	2011	A review proposed to evaluate both treatment for CSF leaks: conservative and operative one. Clinical decisions are taken based on the current literature.
Sonig, A.	2012	This study aims to analyze the risk factors associated with the development of posttraumatic meningitis through the analysis of the NIS database. It turned out that cerebrospinal fluid rhinorrhea and CSF otorrhea are independent predictors of posttraumatic meningitis. The second goal was to analyze the overall hospitalization cost related to posttraumatic meningitis and factors associated with that cost. Meningitis and CSF fistulas resulted as independent risk factors to significantly increased hospitalization cost.
Tebruegge, M.	2008	This review studies recurrent bacterial meningitis and its relations with anatomical anomalies (most common cause) and immunodeficiences. Early diagnosis is fundamental to prevent further episodes.
Durand, M. L.	1993	The authors studied the characteristics of acute bacterial meningitis, including epidemiology, eziology, and mortality.
Adriani, K. S.	2007	A prospective study evaluating episodes of recurrent bacterial meningitis. Remote head injury and CSF leakage are considered as predisposing conditions.
Heckenberg, S. G.	2014	A review of the epidemiology, pathophysiology, and management of bacterial meningitis that shows adequate and prompt treatment of bacterial meningitis (antibacterial) is critical to improve outcomes.
Sağlam, M.	2013	This article studies causative agents of bacterial meningitis by culturing CSF samples. Most frequently isolated agents were H. influenzae, N. meningitidis, and Str. Pneumoniae.
Hernandez, J.L.	2001	An analysis of both clinical and microbiological characteristics of a series of patients with infection by Staphylococcus schleiferi. The results showed the importance of careful identification of Staphylococcus schleiferi in the clinical microbiology laboratory.
Chang, W.N.	2007	A clinical comparison of meningitis caused by S. Aureaus and coagulase-negative Staphylococcus (CoNS). The study revealed an increase of methicillin-resistant, postneurosurgical staphylococcal infection in acute bacterial meningitis. Patients with CoNS infection presented younger age at onset and a lower mortality rate.
Garg, R. K.	2017	The article studies a complication of acute coagulase-negative Staphylococcus (CoNS) meningitis: brainsteam infarct.
Oud, L.	2011	A study about community acquired coagulase-negative Staphylococcus meningitis.
Lin, W.-S.	2013	A case report that underlines the importance of a prompt identification of panspinal epidural abscess with detailed clinical, neurologic, and neuroimaging studies.
Dumas, G.	2012	Subacute otitis media complicated by labyrinthitis, early onset of facial paralysis, or any other oranial nerve palsy should raise suspicion of tuberculosis. The diagnostic workup should include histological and bacteriologic samples, liver markers of intacellular damage, as well as ELlspot test. The prognosis remains poor especially in immunocompromised patients despite appropriate treatment.
Karagol, B.S.	2010	A study that reveales the utmost importance of screening studies in order to be aware of the pathogenic potential of cephalohematomas.
Hedberg, A.	2004	A study about rifampicin and fusidic acid therapy in a patient with severe hypersensitivity reaction to vancomycin.
Van de Beek, D.	2004	A nationwide study in the Netherlands determines clinical features and prognostic factors in adults with community-acquired acute bacterial meningitis: the mortality associated with bacterial meningitis is high, and the strongest risk factors for an unfavorable outcome are those that are indicative of systemic compromise, a low level of consciousness, and infection with *S. pneumoniae*.
Forgie, S. E.	2016	A review that helps the clinician to understand how the history related to sings of meningitis (Kernig, Brudzinski, Amoss) is still germane to clinical practice today.
Mehndiratta, M.	2012	Kernig’s and Brudzinski’s signs are not very sensitive for detecting meningitis and, when absent, should not be inferred as there is no evidence of meningitis. Although the sensitivity is quite low, the high specificity suggests that if Kernig’s or Brudzinski’s sign is present, there is a high likelihood for meningitis. The two signs, Kernig’s and Brudzinski’s, are often performed together in clinical practice.
Nakao, J. H.	2014	A prospective observational study of neurologically intact emergency department (ED) patients undergoing lumbar puncture in two inner city academic EDs to validate the sensitivity and specificity of jolt accentuation (exacerbation of a baseline headache with horizontal rotation of the neck) and to assess the sensitivity and specificity of Kernig sign, Brudzinski sign, and nuchal rigidity, in predicting cerebrospinal fluid (CSF) pleocytosis in individuals being assessed for meningitis.
Busl, K. M.	2017	This article reviews the concept of brain injury-induced immune modulation, and summarizes available data and knowledge on nosocomial meningitis and ventriculitis, and systemic infectious complications in patients with traumatic brain injury, ischemic stroke, intracerebral hemorrhage, subarachnoid hemorrhage, and status epilepticus.
Li, Y. M.	2013	When Staphylococcus lugdunensis is identified, a virulent and prolonged clinical course with the production of destructive lesions, similar to those with S. aureus, should be expected. A course of antibiotic therapy and aggressive management that may include surgical treatment will be needed.
Matas, A.	2015	A case report of a patient who presented with a single, large, right, frontoparietal abscess that evolved despite conventional antibiotic treatment, in the absence of bacteremia and endocarditis. Further studies highlighted the presence of Staphylococcus Ludgunensis.
Noguchi, T.	2018	A case report of Staphylococcus epidermidis meningitis in a patient with neutropenia without a neurosurgical device who was successfully treated.
Bijlsma, M.W.	2016	A study on causative pathogens, clinical characteristics, and outcome of adult community-acquired bacterial meningitis after the introduction of adjunctive dexamethasone treatment and nationwide implementation of paediatric conjugate vaccine.
Uzzan, B.	2006	This study demonstrates that procalcitonin is a good biological diagnostic marker for sepsis, severe sepsis, or septic shock; has difficult diagnoses in critically ill patients; and is superior to C-reactive protein. Procalcitonin should be included in diagnostic guidelines for sepsis and in clinical practice in intensive care units.
Schlenk, F.	2009	This study reveales that a decrease in microdialysis glucose combined with the presence of fever detected bacterial meningitis with acceptable sensitivity and specificity, while CSF chemistry failed to indicate bacterial meningitis. In patients with subarachnoid hemorrhage (SAH), where CSF cell count is not available or helpful, microdialysis might serve as an adjunct criterion for early diagnosis of bacterial meningitis.
Vikse, J.	2015	This study underlines the importance of serum procalcitonin (PCT) as a highly accurate diagnostic test for rapid differentiation between bacterial and viral causes of meningitis in adults.
Khalili, H.	2015	The results of the study indicate that peripheral blood leukocyte count, fever (>38 ˚C), and white blood cells rise (>10%) are non-reliable markers for diagnosis of bacterial meningitis in patients with severe traumatic brain injury (TBI).
Kaabia, N.	2002	This report describes a case of S. lugdunensis meningitis, occurring six days after a endoscopic ventriculostomy, in a 12-year-old child. Coagulase-negative Staphylococcus sp. was isolated in pure culture from the cerebrospinal fluid and was definitely identified as Staphylococcus lugdunensis after the 16S ribosomal DNA gene and rpoB gene were sequenced.
Kastrup, O.	2005	The review summarizes imaging findings and recent advances in the diagnosis of pyogenic brain abscess, ventriculitis, viral disease including exotic and emergent viruses, and opportunistic disease. For each condition, the ensuing therapeutic steps are presented.
Proulx, N.	2005	There is an independent incremental association between delays in administrating antibiotics and mortality from adult acute bacterial meningitis. Inappropriate diagnostic-treatment sequences were significant predictors of such treatment delays.
Brouwer, M. C.	2007	This article shows that corticosteroids significantly reduced hearing loss and neurological sequelae, but did not reduce overall mortality. Data support the use of corticosteroids in patients with bacterial meningitis in high-income countries. We found no beneficial effect in low-income countries.
Zhao, Z.	2019	Third-generation cephalosporin therapy does not have a different prognosis for negative CSF culture of neonatal bacterial meningitis in term infants in this study.
Brouwer, M. C.	2010	This review provides recommendations for empirical antimicrobial and adjunctive treatments for clinical subgroups and review available laboratory methods in making the etiological diagnosis of bacterial meningitis. It also summarizes risk factors, clinical features, and microbiological diagnostics.
Vijayan, P.	2019	Postsurgical device-related meningitis caused by multidrug-resistant coagulase-negative Staphylococci often complicates the treatment options. Monitoring the rational use of linezolid is crucial to avoid the spread of resistance and also comprehensive perioperative care to prevent health care-associated infection.
Organization, W. H.	1998	Practical guidelines that help clinicians in the control of epidemic meningococcal disease.
Jiang, H.	2017	This article aims to investigate the prevalence and antibiotic resistance profiles of cerebrospinal fluid (CSF) pathogens in children with acute bacterial meningitis in Southwest China. Gentamycin, ofloxacin, linezolid, and vancomycin were identified as the most effective antibiotics for Streptococcus pneumoniae, each with susceptibility rates of 100%. It was notable that other emerging pathogens, such as Listeria monocytogenes and group D streptococcus, cannot be underestimated in meningitis.
De Gans, J.	2002	The study demonstrates that early treatment with dexamethasone improves the outcome in adults with acute bacterial meningitis and does not increase the risk of gastrointestinal bleeding.
Wall, E. C.	2018	Osmotic therapies have been proposed as an adjunct to improve mortality and morbidity from bacterial meningitis. Data from trials to date have not demonstrated a benefit on death, but may reduce deafness. Osmotic diuretics, including glycerol, should not be given to adults and children with bacterial meningitis unless as part of a carefully conducted randomised controlled trial.
Ratilal, B. O.	2015	A study carried out to evaluate the effectiveness of prophylactic antibiotics for preventing meningitis in patients with basilar skull fractures. The evidence does not support prophylactic antibiotic use in patients with basilar skull fractures, whether or not there is evidence of CSF leakage. Until more research is available, the effectiveness of antibiotics in patients with basilar skull fractures cannot be determined because studies published to date are flawed by biases.
Gianella, S.	2006	A case of cerebral abscess as an embolic complication of infective endocarditis owing to S. Lugdunensis. Conservative methods were used and the efficacy of this approach is supported by the results of the literature review. The described treatment validates the thesis that, in select clinical settings, it is possible to cure such a serious disorder without surgical intervention.
Rebai, L.	2019	This article reveals that linezolid is an alternative to vancomycin for the treatment of postneurosurgical infection (PNSI) caused by methicillin-resistant *Staphylococcus* (MRS) with a high rate of efficacy.
Denetclaw, T. H.	2014	This case report shows the efficacy of low-volume intrathecal daptomycin in treatment of ventriculostomy-associated meningitis caused by multidrug resistant coagulase-negative staphylococcus epidermidis.
Vena, A.	2013	This article shows the efficacy of daptomycin plus trimethoprim/sulfamethoxazole combination therapy in post-neurosurgical meningitis caused by linezolid resistant staphylococcus epidermidis.
Jiang, H.	2013	The detected pathogens that cause bacterial meningitis to develop high resistance to commonly used antibiotics. To prevent misdiagnosis, careful attention should be paid to bacterial meningitis caused by Cryptococcus neoformans.
Watanabe, S.	2013	Linezolid may be a treatment option for neonates and infants for drain-associated meningitis caused by methicillin resistant Staphylococcus epidermidis.
Lucas, M. J.,	2016	Neurological sequelae occur in a substantial amount of patients following bacterial meningitis. Most frequently reported sequelae are focal neurological deficits, hearing loss, cognitive impairment, and epilepsy.
Crawford, C.	1994	A case report of a bacterial meningitis (owing to H. Influenzae) in a 40-year-old patient that suffered an head injury at 3 years old. This study sets the longest recorded interval between head injury and meningitis.
Okada, J.	1991	Two cases of acute meningitis and cerebrospinal fluid rhinorrhea, in which the head trauma responsible occurred 10 and 30 years before, are presented. The causes of this late onset cannot be clearly explained.
Plaisier, B. R.	2005	A retrospective analysis of patients with post-traumatic meningitis that reveals that admission Glasgow Coma Scale was predictive of good functional outcome, but it plays no role in death prediction.
Lai, W.-A.	2011	This study shows that he relative frequency of implicated pathogens of elderly acute bacterial meningitis (ABM) is similar to that of non-elderly ABM. Compared with non-elderly patients, the elderly ABM patients have a significantly lower incidence of peripheral blood leukocytosis. The mortality rate of elderly ABM remains high, and the presence of shock and seizures represents important prognostic factors.
Tian, R.	2015	This article analyzes the epidemilogy of post-neurosurgical meningitis in the northern mainland of China. Post-neurosurgical meningitis usually occurs in the autumn and winter of the year. Gram-positive organisms, which are sensitive to compound sulfamethoxazole and vancomycin, are the most common causative pathogens of post-neurosurgical meningitis.
Drinkovic, D.	2002	Two cases report that describe the onset of coagulase negative Stafilococci meningitis in neonates without CSF shunts. The succesful therapy consisted of vancomycin and rifampicin.
Tian, L.	2019	A retrospective study based on analysis of samples from patients with CNS infection in a clinical microbiology laboratory at a teaching hospital in China over a 6-year period indicated that the most common etiological agents were the bacteria Acinetobacter Baumani and Staphylococcus Aureus. The antibiotic resistance levels of A. Baumanni were found to be high and of concern, whereas isolates of C. neoformans were found to be sensitive to antifungal antibiotics.
AlDhaleei, W. A.	2019	Case report of a S. Lugdunensis endocarditis complicated by both embolic stroke and meningitis.
Sasaki, Y.	2016	Case report of a 51-year-old man undergone surgery forn Rathke’s cleft cyst complicated with CSF rinhorrea. CSF microbiological findings led to a diagnosis of bacterial meningitis.
Chitnis, A. S.	2012	This study suggests hygienic and behaviral norms to decrease the rate of bacterial meningitis transmitted by health care personnel
Kawaguchi, Y.	2010	A case report of a postoperative meningitis in a patient with cervical cord tumor, treated with intravenous linozelid. This latter drugs were shown to be one of the first choices in methicillin-resistant *Staphylococcus epidermidis* (MRSE) meningitis.
Nagashima, G.	2008	A case report of postneurosurgical meningitis caused by methicillin-resistant Staphylococcus epidermidis. The patient was succesfully treated with Linezolid.
Stevens, N. T.	2008	This article shows the full role of biofilm in Staphylococcus epidermidis meningitis.
Huang, C.R.	2005	This article analyzes the clinical characteristics and therapeutic outcomes of adult meningitis caused by coagulase-negative staphylococci (CoNS). Epidermidis was the most common CoNS subtype involved. All involved CoNS strains were oxacillin resistant. The therapeutic result showed that adult CoNS meningitis had a high mortality rate.
Neri, M.	2018	This study analyzes the expression of Aquaporin-4 in fatal traumatic brain injury. Further studies evaluated the correlation with cluster differentiation (CD)68, ionized calcium binding adapter molecule 1 (IBA-1), hypoxia induced-factor 1α (HIF-1α), glial fibrillary acid protein (GFAP), and CD15.
La Russa R.	2019	This study evaluated 56 experimental studies for diagnostic usefulness of specific immunohistochemical assays in the diagnosis of sepsis as a cause of death.
Maiese A.	2019	In this article, for the first time, the usefulness of Procalcitonin as a specific target for immunohistochemical assays was shown, investigating the implementation of such a test for forensic purposes in different organs.
Maiese A.	2017	In this immunohistochemical study, s-TREM-1 antibodies are used. The findings indicate that immunohistochemical assays for s-TREM-1 in sections from multiple organ samples (brain, heart, lung, liver, and kidney) could enable post-mortem diagnosis of sepsis with good sensitivity and specificity.

**Table 2 ijms-21-04148-t002:** Semi-quantitative evaluation of immunohistochemical reaction to specific markers in brain samples with correlate timing.

Biomarkers	Post-Traumatic Interval /Time of Death After Trauma					
	Almost Immediate	1 Day (20 ± 6 h)	3 Days (72 ± 10 h)	7 Days (6 ± 2 Days)	14 Days (14 ± 4 Days)	30 Days (30 ± 10 Days)
AQP4	+/-	+++	+++	++++	++++	++++
HIF-1α	+/-	++	+++	+++	++	++
GFAP	+/-	++	+++	+++	++++	++++
CD68	+/-	+	+++	+++	++++	++++
IBA-1	+/-	++	+++	+++	+++	+++
CD-15	+/-	++	+++	+++	++	++

Thirty days after trauma. Antibodies for the following markers were adopted: AQP4 (aquaporin 4); HIF-1α (hypoxia induced-factor 1α); GFAP (glial fibrillary acid protein); CD68 (macrophage cluster differentiation 68); IBA-1 (ionized calcium binding adapter molecule 1); and CD-15 (neutrophilic cluster differentiation 15). The intensity of molecular expression was assessed using a semi-quantitative method: +/- represents a low expression. + depicts a mild expression. ++, +++, and ++++ symbolize an increasing degree of expression up to the majority of all cells, respectively.

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
