# Peer review of "Post-Traumatic Meningitis Is a Diagnostic Challenging Time: A Systematic Review Focusing on Clinical and Pathological Features"

_ijms, 2020, doi:10.3390/ijms21114148_

Round 1
Reviewer 1 Report
1) You mention that ‘the appearance of CSF leakage should always arouse suspicion of meningitis’, however meningitis is the most significant risk associated with cranial CSF leaks. There is no increased risk of meningitis with a spinal CSF leak. Perhaps, you could make that distinction clear somewhere.
2) In Figure 1, remove the purple PRISMA logo and the name (PRISMA Flow Diagram) and instead make the figure caption more descriptive.
3) In the ‘Risk of Bias’ you mention that the accuracy of clinical procedures has changed over the 59 years. Perhaps one of your ‘search criteria’ should have been a time limit on the articles considered. In other words, you could have made an overview of these clinical procedures, demonstrate how they changed over the years and then decide to include papers only from the last time period where the procedures are most relevant. I do not wish you to now change your paper too much, but you still could include a subsection where you describe the clinical procedures and how they changed.
4) Please be more specific about why you excluded up to 218 full-text articles based on their ‘study design’.
5) Defend why papers with ‘not explicit mention of ethical issue’ should be excluded.
6) You define the acronym PRISMA twice (First in row 21 which is the abstract, and then again in row 57). But, the acronym CSF is only defined in row 114 while used in the abstract without definition.
7) In row 289, you use ‘cerebrospinal fluid’ while the CSF acronym has been defined before.
8) In row 297, you start, ‘When CSF cannot be obtained…’. Can you mention a few examples of when/why that might be the case?
9) I suggest you use titled subsubsections to separate some of your paragraphs from others.
Author Response
We appreciate the comments of the reviewer and have made revisions accordingly. Changes were fully incorporated in the appropriate sections of the revised manuscript. We thank the reviewer for the kind suggestions.
Reviewer 1
1) You mention that ‘the appearance of CSF leakage should always arouse suspicion of meningitis’, however meningitis is the most significant risk associated with cranial CSF leaks. There is no increased risk of meningitis with a spinal CSF leak. Perhaps, you could make that distinction clear somewhere.
We thank the Reviewer for pointing out this inaccuracy. Considering that the association between meningitis and cranial CSF leaks (manifesting as rhinorrhea or otorrhea) is widely illustrated in the “Epidemiology and risk factors” section, when dealing with the role of skull fractures, we have specified “cranial CSF leakages (manifesting as rhinorrhea or otorrhea)” in both the abstract and the conclusions (indeed, we are afraid that a comparison between cranial and spinal CSF leak might further complicate the topic of the review and be confounding for the readers).
2) In Figure 1, remove the purple PRISMA logo and the name (PRISMA Flow Diagram) and instead make the figure caption more descriptive.
We did it.
3) In the ‘Risk of Bias’ you mention that the accuracy of clinical procedures has changed over the 59 years. Perhaps one of your ‘search criteria’ should have been a time limit on the articles considered. In other words, you could have made an overview of these clinical procedures, demonstrate how they changed over the years and then decide to include papers only from the last time period where the procedures are most relevant. I do not wish you to now change your paper too much, but you still could include a subsection where you describe the clinical procedures and how they changed.
We modified the “Diagnostic approach” and the “Therapeutic management” paragraphs and into the subsections 4.4.2 Surrogate serum markers; 4.4.4 Novel diagnostic tools; 4.5.1 Antibiotic and steroid treatment in post-traumatic meningitis and 4.5.2 Antibiotic prophylaxis after TBI: a matter of debate, we summarized the clinical procedures and how they changed in the 59-years window of time. Especially, you’ll read paragraph 4.6.
4) Please be more specific about why you excluded up to 218 full-text articles based on their ‘study design’.
We specified that “This search identified 321 articles, which were then screened based on their abstract to identify their relevance in respect
• of the human study, so we excluded animal studies,
• clinical features,
• diagnosis,
• post-mortem findings,
• management of the study so we excluded methodologically incomplete design studies and with no explicit mention about ethical issues;
A total of 81 studies fulfilled the inclusion criteria”.
5) Defend why papers with ‘not explicit mention of ethical issue’ should be excluded.
The following phrase has been added in the text. “With regard to ethical issues, we have discarded the papers where the obtaining of informed consent on patients was not mentioned, where the approval of the ethics committee was not found and, finally, where the permission to publish personal data was not clearly explicit”.
6) You define the acronym PRISMA twice (First in row 21 which is the abstract, and then again in row 57). But, the acronym CSF is only defined in row 114 while used in the abstract without definition.
We modified the text; correction was made in the abstract and on row 57. The same we made in the abstract and on row 130 (114 in the first version).
7) In row 289, you use ‘cerebrospinal fluid’ while the CSF acronym has been defined before.
We modified the text.
8) In row 297, you start, ‘When CSF cannot be obtained…’. Can you mention a few examples of when/why that might be the case?
Although CSF analysis is essential for the etiological diagnosis of meningitis, in some circumstances the lumbar puncture might be extremely difficult - as it happens in cases of serious spine deformities, previous back surgery, obesity- or highly discouraged, for instance when the patient is at risk for developing intracranial hypertension, that could be aggravated by the procedure, or when he/she is receiving concomitant anticoagulant therapy. In other (rare) cases, although the spinal needle is correctly placed, no CSF can be actually drained (the so-called “dry spinal tap” or “punctio sicca”) due to dehydration. The text has been clarified as follows: “When CSF cannot be obtained (e.g. due to the high risk of intracranial hypertension, or for difficulties in performing the lumbar puncture related to the spine anatomy, prior back surgery, obesity etc) surrogate serum markers supporting the clinical suspicion of meningitis are needed”.
9) I suggest you use titled subsubsections to separate some of your paragraphs from others.
Following the Reviewer’s advice, the “Diagnostic approach” and the “Therapeutic management” paragraphs have been divided into subsections (4.4.1 CSF analysis; 4.4.2 Surrogate serum markers; 4.4.3 The role of neuroimaging in meningitis; 4.4.4 Novel diagnostic tools; and 4.5.1 Antibiotic and steroid treatment in post-traumatic meningitis; 4.5.2 Antibiotic prophylaxis after TBI: a matter of debate; 4.5.3 The management of CSF leaks).
Reviewer 2 Report
Comments:
- Was this systematic review registered on PROSPERO? If the response is not, I suggest to the author that the next systematic review registers on PROSPERO. https://www.crd.york.ac.uk/prospero/
- I could not find the PICO question on the manuscript. I suggest adding the hypothesis and the PICO question of this systematic review (on the abstract and the manuscript).
- I couldn't find a table with the results of 81 articles included in this manuscript. I suggest that the author add a table with details of all studies included in this systematic review.
- In the abstract, the authors included the word meta-analyses. "The present systematic review was carried out according to the Preferred Reporting Items for Systematic Review and Meta-Analyses (PRISMA) standards." Was a meta-analysis performed? If not, please remove the word "meta-analysis" from the abstract.
- Risk of Bias. I could not find the table with the study's evaluation. Please add the quality of assessment (suggestion: Newcastle-Ottawa, Whiting P, Harbord R, Kleijnen J. BMC medical research methodology 2005; 5: 19., etc.).
- It is also possible to perform immunohistochemical studies for the postmortem diagnosis of sepsis-related to meningitis. How these markers (AQP4, HIF-1α, GFAP, CD68, IBA-1, CD-15) can identify sepsis-related to meningitis?
- Conclusion: "Our data indicate that miR-21, miR575 92 and miR-16 have high predictive power in discriminating trauma brain-cases from controls and could represent promising biomarkers as a strong predictor of survival, and useful tool for postmortem diagnosis of traumatic brain injury." Is this data from the research group? How can be evaluated the miR-21 and -16 as a predictor of survival?
- Page 5, line 167. Please add a full word to the blood-brain barrier after that use the abbreviation BBB.
Author Response
We appreciate the comments of the reviewer and have made revisions accordingly. Changes were fully incorporated in the appropriate sections of the revised manuscript. We thank the reviewer for the kind suggestions.
Reviewer 2
1) Was this systematic review registered on PROSPERO? If the response is not, I suggest to the author that the next systematic review registers on PROSPERO. https://www.crd.york.ac.uk/prospero/
Thank you for this suggestion; we’ll follow your comment.
2) I could not find the PICO question on the manuscript. I suggest adding the hypothesis and the PICO question of this systematic review (on the abstract and the manuscript).
Thank you for this comment. We added the hypothesis and the PICO question of this systematic review on the abstract and the manuscript.
3) I couldn't find a table with the results of 81 articles included in this manuscript. I suggest that the author add a table with details of all studies included in this systematic review.
The table has been added in the text. It is a very long table…
4) In the abstract, the authors included the word meta-analyses. "The present systematic review was carried out according to the Preferred Reporting Items for Systematic Review and Meta-Analyses (PRISMA) standards." Was a meta-analysis performed? If not, please remove the word "meta-analysis" from the abstract.
We removed the word meta-analysis from the abstract.
5) Risk of Bias. I could not find the table with the study's evaluation. Please add the quality of assessment (suggestion: Newcastle-Ottawa, Whiting P, Harbord R, Kleijnen J. BMC medical research methodology 2005; 5: 19., etc.).
We specified that “This search identified 321 articles, which were then screened based on their abstract to identify their relevance in respect
• of the human study so we excluded animal studies,
• clinical features,
• diagnosis,
• post-mortem findings,
• management of the study so we excluded methodologically incomplete design studies and with no explicit mention about ethical issues;
A total of 81 studies fulfilled the inclusion criteria”.
About study’s evaluation, we modified the “Diagnostic approach” and the “Therapeutic management” paragraphs and into subsections 4.4.2 Surrogate serum markers; 4.4.4 Novel diagnostic tools; 4.5.1 Antibiotic and steroid treatment in post-traumatic meningitis and 4.5.2 Antibiotic prophylaxis after TBI: a matter of debate, we summarized the clinical procedures and how they changed in the 59-years window of time.
6) It is also possible to perform immunohistochemical studies for the postmortem diagnosis of sepsis-related to meningitis. How these markers (AQP4, HIF-1α, GFAP, CD68, IBA-1, CD-15) can identify sepsis-related to meningitis?
Thank you for this comment. In the text we specified: “As far as clinical features are concerned, post-traumatic meningitis does not differ from community-acquired cases: however, awareness impairment is generally more severe, and the clinical picture might be clouded by the presence of other brain lesions (e.g. intracranial bleeding, cerebral edema) or concomitant medical conditions, which makes the early recognition of meningitis even more challenging”.
So, consequently, in the text we added the phrase: “While evaluating brain injury several proteins such as Aquaporin4 (AQP4), CD58, CD68 e Antibodies anti-glial fibrillary acid protein (GFAP) should be researched. Previous studies identified that these molecular markers expression show a temporal correlation with the event in case of traumatic brain injury… It is also possible to perform immunohistochemical studies for the post-mortem diagnosis of sepsis related to meningitis”.
7) Conclusion: "Our data indicate that miR-21, miR575 92 and miR-16 have high predictive power in discriminating trauma brain-cases from controls and could represent promising biomarkers as a strong predictor of survival, and useful tool for postmortem diagnosis of traumatic brain injury." Is this data from the research group? How can be evaluated the miR-21 and -16 as a predictor of survival?
We modified the text and we have expanded this paragraph to better specify the comment raised by the reviewer.
8) Page 5, line 167. Please add a full word to the blood-brain barrier after that use the abbreviation BBB.
We did it.